# Phosphoregulated orthogonal signal transduction in mammalian cells

Leo Scheller [1,3], Marc Schmollack[1,4], Adrian Bertschi [1], Maysam Mansouri[1], Pratik Saxena[1] & Martin Fussenegger [1,2 ✉]

Orthogonal tools for controlling protein function by post-translational modifications open up new possibilities for protein circuit engineering in synthetic biology. Phosphoregulation is a key mechanism of signal processing in all kingdoms of life, but tools to control the involved processes are very limited. Here, we repurpose components of bacterial two-component systems (TCSs) for chemically induced phosphotransfer in mammalian cells. TCSs are the most abundant multi-component signal-processing units in bacteria, but are not found in the animal kingdom. The presented phosphoregulated orthogonal signal transduction (POST) system uses induced nanobody dimerization to regulate the trans-autophosphorylation activity of engineered histidine kinases. Engineered response regulators use the phospho-histidine residue as a substrate to autophosphorylate an aspartate residue, inducing their own homodimerization. We verify this approach by demonstrating control of gene expression with engineered, dimerization-dependent transcription factors and propose a phosphoregulated relay system of protein dimerization as a basic building block for next-generation protein circuits.

[1] Department of Biosystems Science and Engineering, ETH Zurich, Mattenstrasse 26, CH-4058 Basel, Switzerland. [2] University of Basel, Faculty of Science, Mattenstrasse 26, CH-4058 Basel, Switzerland. [3] Present address: Institute of Bioengineering, École Polytechnique Fédérale de Lausanne, Lausanne CH-1015, Switzerland. [4] Present address: Microbial Biotechnology, Campus Straubing for Biotechnology and Sustainability, Technical University of Munich, Straubing DE-94315, Germany. ✉email: martin.fussenegger@bsse.ethz.ch

I ntracellular protein circuit engineering promises increased control over cellular behavior. Technologically, however, much progress must be made in developing suitable systems to achieve this goal. In nature, highly complex signaling networks have evolved to regulate diverse functions in living cells. These functions include the actions of auxiliary regulators[1], tuning of signal detection windows[2], controlling on/off kinetics[3,4], enabling dynamic effects based on fast negative feedback[5], and modulating noise resistance and ultrasensitivity[6,7]. Synthetic biology aims to capitalize on these sophisticated control systems by transferring pathways from one species to another, by engineering and rerouting existing pathways, or even by designing new pathways altogether. These engineered systems have the potential to find applications in medicine and biotechnology, as well as contributing to the understanding of the principles that underlie natural signaling by offering a synthetic platform for dissecting the interplay and functions of individual signaling proteins[8–12]. The development of new platforms for protein circuit design aids in these efforts and in particular modular systems that are compatible with previous designs may enable the construction of higher order circuits.

In contrast to single-step input–output systems, such as gene switches based on the TetR protein family[13], the focus of engineering protein circuits lies on switchable protein–protein interactions (PPis) that require post-translational protein modifications (PTMs). Most current designs for synthetic PTMs rely on protein cleavage with viral proteases. Controlling protein function by proteolytic cleavage enables the construction of a very diverse set of systems for different applications[14–19]. All of these approaches greatly profit from the modularity and specificity of viral proteases and signaling networks based on proteases can recapitulate some of the effects that distinguish protein networks from gene networks. However, regulating protease activity remains challenging and the irreversibility of proteolysis limits the versatility of this approach.

Most natural protein circuits are based on the rapid control of reversible PTMs, and in particular, control of protein function by phosphorylation. In this context, engineered systems for controlling protein function by phosphoregulation represent a very attractive target[20]. Pioneering studies include the design of phosphorylation-dependent nuclear localization sequences and engineered phosphoregulated protein interaction to control cellular functions[21–24]. A modular system would enable a more fundamental redesign of such pathways and would help us to elucidate why specific pathway architectures are favored by nature. The main reasons why such systems have remained largely elusive so far are presumably the issues of crosstalk with existing pathways and the lack of modular tools that can be selectively activated[8,25]. A key challenge for controlling protein function for uses in synthetic biology is to engineer protein switches that are controlled by small molecules[26–29]. In subsequent steps, such protein switches may then be integrated in protein circuit design.

Here, we aim to characterize and develop functional modules to integrate orthogonal phosphoregulated protein switches into the synthetic biology toolbox. We propose a phosphoregulated relay system of protein dimerization as a basic building block for such circuits. Induced protein dimerization is widely used in other orthogonal systems, such as split transcription factor approaches based on dCas9, Gal4, or TetR[30–32]. Phosphoregulated protein dimerization could complement these systems by providing an additional layer of control. In this work, we engineer the bacterial histidine kinases (HKs) DcuS, EnvZ, and NarX to be activated by the small molecule caffeine. We generate cytosolic versions of these proteins and test kinase activity by response regulator (RR)- mediated reporter gene expression in mammalian cells. These experiments serve as a proof of concept

for basic orthogonal phosphoregulated signal transduction. We demonstrate the utility of HK/RR pairs for generating a phosphoregulated relay system of protein dimerization, which is compatible with activating gene expression with a split transcription factor. These components may be integrated into the next generation of synthetic protein circuits that recapitulate some of the versatility of native signaling pathways.

## Results

**Design strategy**. The design is informed by the function of native bacterial two-component systems (TCSs), which rely on the trans-autophosphorylation of a histidine residue in the HK to form phosphohistidine. The phosphohistidine serves as substrate for the subsequent phosphotransfer to an aspartate residue in the RR to form phosphoaspartate. As a starting point, we chose the prototypical homodimeric *Escherichia coli* HK DcuS, which senses C4-dicarboxylates (e.g. fumarate) and controls the RR DcuR[33] (Fig. 1a). When these proteins are expressed in mammalian cells, phosphotransfer between DcuS and DcuR is constitutively active[34]. The cytoplasmic domain of DcuS (DcuS$_{203–543}$; numbering according to UniProt[35] ID P0AEC8) contains an N-terminal Per-Arnt-Sim (PAS) domain connected by a linker, whose function is incompletely understood, to a kinase domain consisting of the Dimerization Histidine phosphotransfer (DHp) and Catalytic ATP-binding (CA) domains. The kinase trans-autophosphorylates its DHp domain upon receptor activation. The PAS domain and the linker between the PAS and kinase domains are located at the dimer interface between the receptor chains and play a major role in regulating kinase activity[36–38] (Fig. 1a). Our first aim was to generate N-terminal truncation mutants to identify minimal domains of DcuS that are not active when expressed cytosolically, but retain intact kinase domains (Fig. 1b). We then planned to fuse these minimal domains to an anti-caffeine heavy chain nanobody (acV$_H$H) for chemically induced dimerization (CiD)[39,40] (Fig. 1c).

We hypothesized that CiD of the kinase domain of a bacterial HK would trigger trans-autophosphorylation of the homodimer, followed by phosphotransfer to (and dimerization of) the corresponding RR (Fig. 1d). To test system function, we fused the RR to a transactivator domain and monitored phosphorylation-dependent RR dimerization by measuring reporter gene expression from an inducible promoter, activated by the binding of the dimerized RR. The phosphotransfer is expected to be specific for the engineered HK/RR pair, as they have coevolved for specific binding interfaces[41–44]. The regulatory domain of the RR contains a catalytic center that uses phosphohistidine as a substrate for autophosphorylation[45]. Phosphotransfer between HK and RR therefore consists of two autophosphorylation events, which reduces the likelihood of off-target phosphorylation. It should be noted that mammalian orthologues to bacterial TCSs have not been identified, and HK/RR pairs are not present in the animal kingdom[41,42]. Therefore, we refer to the engineered kinase as the orthogonal receptor kinase (ORK; fusion of acV$_H$H to the truncation mutant of DcuS; Fig. 1e) and the RR as the orthogonal gene expression regulator (OGR; fusion of VP16 to DcuR; Fig. 1e), and we designate the full system (ORK/OGR/reporter) as the phosphoregulated orthogonal signal transduction (POST) system.

**Defining the DcuS minimal kinase domain**. We compared the kinase activity of several N-terminally truncated variants of the DcuS intracellular domain (Fig. 2a, b). The DcuS N-terminal truncation variants were cytosolically expressed in HEK-293T cells, together with the response regulator DcuR fused to the transactivator VP16 (orthogonal gene expression regulator; OGR;

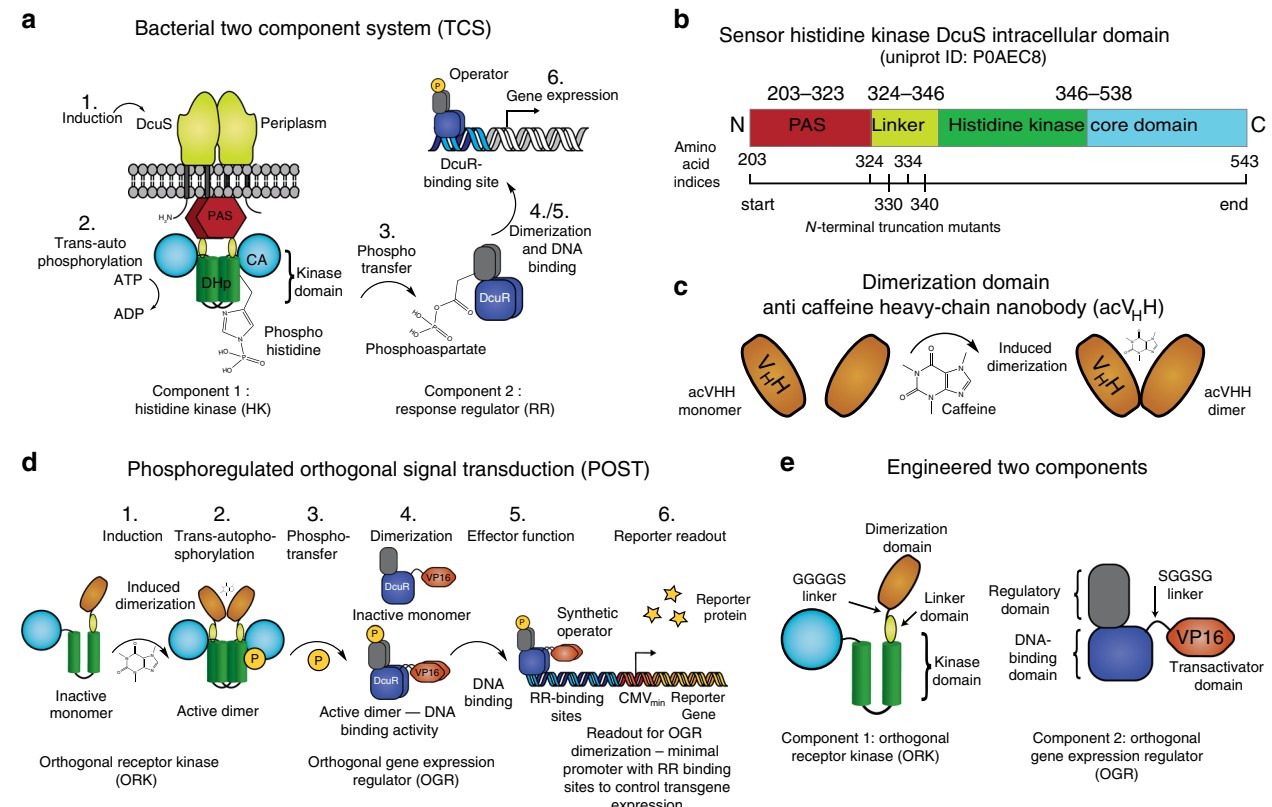

**Fig. 1 Design of the phosphoregulated orthogonal signal transduction (POST) system. a** Mechanism of the native signal cascade activated by the bacterial histidine kinase DcuS. (1) Upon activation, (2) the homodimeric histidine kinase DcuS trans-autophosphorylates a histidine residue in its kinase domain, consisting of the dimerization and histidine-containing phosphotransfer (DHp) domain and the catalytic and ATP-binding (CA) domain. (3) This phosphohistidine is the substrate for the response regulator DcuR that catalyzes the autophosphorylation of an aspartate residue in its dimerization domain. (4) Phosphorylated DcuR dimerizes and (5) binds response elements in its operator site to control (6) gene expression. **b** Linear schematic depiction of the N-terminal truncation constructs. The constructs start with the amino acid number indicated to the left and end with amino acid number 543 (numbered according to UniProt ID: P0AEC8). **c** The camelid heavy chain nanobody acV$_H$H dimerizes in the presence of caffeine. **d** Schematic illustration of the POST system design. (1) Caffeine induces dimerization of acV$_H$H domains in the engineered orthogonal receptor kinase (ORK), causing (2) kinase trans autophosphorylation and (3) phosphotransfer to an engineered effector protein, such as the orthogonal gene expression regulator (OGR). (4) The effector dimerizes upon phosphotransfer to perform its function, i.e., DNA binding (5), leading to (6) activation of gene expression. **e** Detailed design of ORK and OGR proteins. The regulatory domain catalyzes the transfer of the phosphoryl group from phosphohistidine to one of its own aspartate residues and subsequently dimerizes.

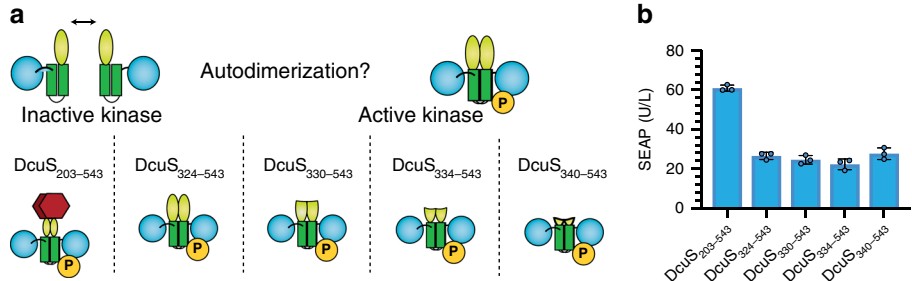

**Fig. 2 Reducing DcuR autodimerization by generating truncation mutants. a** Schematic of DcuS truncation mutants. **b** Reporter gene expression in response to expression of different DcuS N-terminal truncation variants together with DcuR-VP16. The bar chart shows the mean ± s.d. of $n = 3$ biologically independent samples overlaid with a scatter dot plot of the original data points, measured 24 h after transfection, and the results are representative of three independent experiments. Source data are provided as a Source Data file.

$P_{SV40}$-DcuR-VP16-pA). To quantify DcuS activity, we designed a reporter plasmid containing DcuR-specific-binding sites[34] (DcuR response elements; RE) upstream of a minimal promoter that drives expression of the reporter protein, secreted alkaline phosphatase (SEAP; DcuR-RE$_{8x}$-P$_{hCMVmin}$-SEAP-pA; Fig. 1d). The complete intracellular domain of DcuS ($P_{SV40}$-DcuS$_{203-543}$-pA) led

to high reporter gene expression. Variants lacking the PAS domain ($P_{SV40}$-DcuS$_{324-543}$-pA, $P_{SV40}$-DcuS$_{330-543}$-pA, $P_{SV40}$-DcuS$_{334-543}$-pA, $P_{SV40}$-DcuS$_{340-543}$-pA) had reduced activity (Fig. 2b). The basal reporter activity can likely be attributed to a combination of promoter leakiness, residual DcuS dimerization, and uninduced DcuR binding to its response elements. These

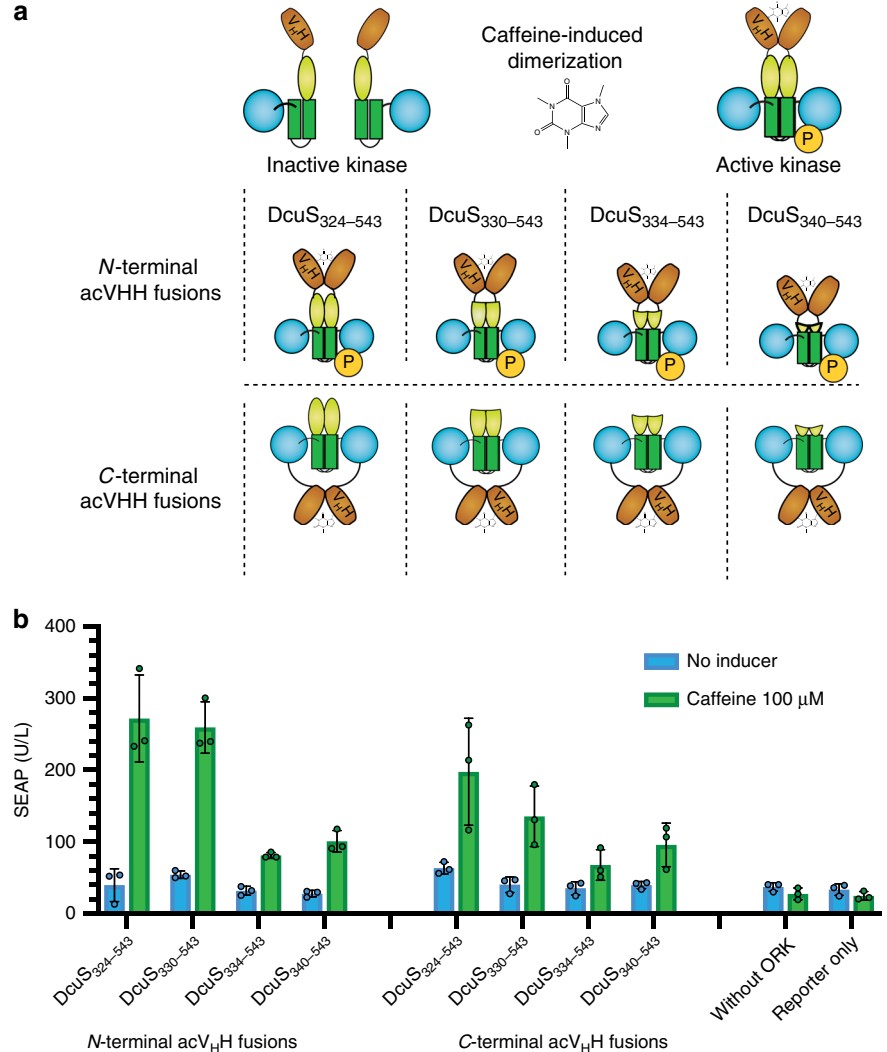

**Fig. 3 Functional comparison of different ORK designs. a** ORKs consisting of different N-terminal truncation variants of DcuS fused N-terminally or C-terminally to the nanobody acV$_H$H, which dimerizes in the presence of caffeine. **b** Reporter gene expression in response to caffeine for POST with N-terminal or C-terminal acV$_H$H fusion ORKs, co-expressed with the OGR (DcuR-VP16). The bar chart shows the mean ± s.d. of $n = 3$ biologically independent samples overlaid with a scatter dot plot of the original data points, measured 24 h after induction and the results are representative of three independent experiments. Source data are provided as a Source Data file.

parameters may be improved in future studies by identifying and optimizing minimal promoters and response elements, and by mutagenesis of DcuR and DcuS.

**Generating caffeine-inducible HKs**. We excluded the constitutively active variant DcuS$_{203–543}$ from further analysis and examined whether CiD could restore the activity of DcuS$_{324–543}$, DcuS$_{330–543}$, DcuS$_{334–543}$, and DcuS$_{340–543}$. We fused these truncated variants via a (GGGGS)-linker to the nanobody acV$_H$H, which dimerizes in the presence of caffeine[39,40]. The N-terminal ($P_{SV40}$-acV$_H$H-DcuS$_{x-543}$-pA;) and C-terminal ($P_{SV40}$-DcuS$_{x-543}$-acV$_H$H-pA) fusions of acV$_H$H to truncated DcuS intracellular domain variants expressed together with the OGR DcuR-VP16 in HEK-293T cells induced reporter gene expression in response to caffeine. Cells transfected with the reporter plasmid alone or with the reporter plasmid and the expression plasmid for the OGR but not for ORKs did not respond to caffeine (Fig. 3a, b). The POST system containing ORKs with DcuS$_{324–543}$ or DcuS$_{330–543}$ had the highest signal-to-noise ratio of induced vs. noninduced SEAP expression under the tested conditions; these constructs were used for further analysis.

We also fused the dimerization domains FRB/FKBP via a (GGGGS)-linker to the identified minimal domain of DcuS ($P_{SV40}$-FRB-DcuS$_{324-543}$-pA; $P_{SV40}$-FKBP-DcuS$_{324-543}$-pA) to test if the POST design strategy can be extended to other CiD systems. FRB/FKBP ORKs together with the OGR robustly induced SEAP expression from the reporter plasmid. We confirmed that FRB ORKs alone, used as the negative control, did not respond to rapamycin (Supplementary Fig. 1).

While the signal-to-noise ratio and sensitivity of the system are not superior to those of endogenous signaling cascades[40], these results indicate the successful implementation of an inducible phosphorylation-dependent protein switch based on the phosphorylation of histidine and aspartate residues.

**Using POST as a dimerization relay system**. Next, we sought to use the phosphorylation-dependent dimerization mechanism of DcuR to establish OGRs as a modular phosphorylation-dependent effector. A phosphoregulated relay system of protein dimerization can serve as a basic building block that is compatible with other synthetic biology systems, such as split transcription factors (Fig. 4a). Phosphorylated DcuR forms homodimers and

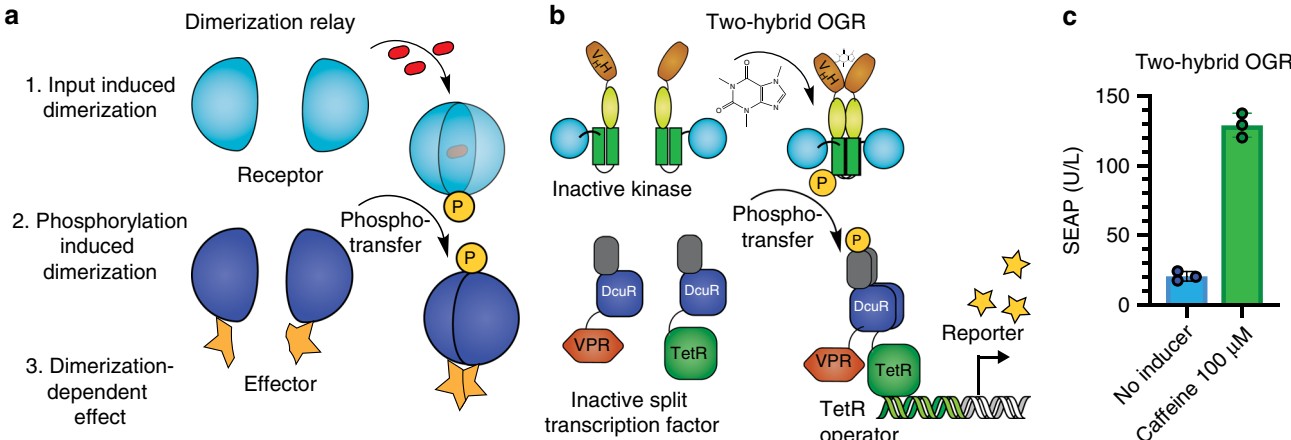

**Fig. 4 POST as a dimerization relay system. a** Schematic of the proposed basic building block of protein circuits using a phosphoregulated relay system of protein dimerization. This system consists of a dimerization-dependent receptor and a dimerization-dependent effector. **b** Schematic of POST as a dimerization relay system by a two-hybrid OGR design. The two-hybrid OGR serves as an example of a phosphoregulated effector protein that performs its function upon dimerization. It consists of DcuR fused to TetR, and DcuR fused to the DNA transactivation domain VPR. Phosphoregulation by ORKs controls the dimerization of TetR-DcuR and DcuR-VPR and only dimerized TetR-DcuR/DcuR-VPR will induce reporter gene expression from a TetR operator. **c** Reporter gene expression from a TetR operator in response to caffeine in the two-hybrid OGR design. The bar chart shows the mean ± s.d. of $n = 3$ biologically independent samples overlaid with a scatter dot plot of the original data points, measured at 24 h after induction, and the results are representative of three independent experiments. Source data are provided as a Source Data file.

subsequently binds to its response elements[46]. DcuR with a mutated phosphoaspartate cannot be induced to bind DNA[47]. To confirm that dimerization and not another phosphorylation-mediated effect is the primary driver of DcuR binding and reporter activation, we generated an acV$_H$H-DcuR fusion protein ($P_{SV40}$-acV$_H$H-DcuR-VP16-pA). This construct triggered transgene expression from the reporter plasmid in response to caffeine without the need for the ORK (Supplementary Fig. 2). To establish POST as a dimerization relay system, we fused DcuR to the DNA-binding protein TetR ($P_{hCMV}$-DcuR-TetR-pA) and to the strong transactivation domain VPR ($P_{hCMV}$-NLS-DcuR-VPR-pA). We reasoned that ORK-mediated DcuR phosphorylation and subsequent dimerization should recruit VPR to the operator and activate gene expression (two-hybrid OGR; Fig. 4b). The strong CMV promoter ($P_{hCMV}$) was used to enhance ORK/two-hybrid-OGR expression levels, and nuclear localization sequences (NLS) were included for DcuR-VPR and the ORK ($P_{hCMV}$-NLS-acV$_H$H-DcuS$_{324-543}$-pA). The functionality of the POST system consisting of ORK/two-hybrid-OGR activation was tested by reporter gene expression from a TetR reporter plasmid ($TetO_7$-$P_{hCMVmin}$-SEAP-pA). The two-hybrid-OGR POST system was responsive to induction with caffeine, consistent with phosphorylation-dependent OGR dimerization (Fig. 4c). Dimerization is a ubiquitous control mechanism to control protein function in both natural and engineered pathways[48]. Thus, a phosphoregulated dimerization mechanism has the potential to be used in various combinations of synthetic and endogenous signaling cascades, as well as for the activation of split proteins to enable diverse functions.

**Dose–response relationships of different POST designs**. We measured dose–response curves of POST systems for ORKs containing N-terminal or C-terminal fusions of the dimerization domain acV$_H$H with DcuS$_{324-543}$ or DcuS$_{330-543}$ together with the DcuR-VP16 OGR. All the POST systems were inducible with a similar signal-to-noise ratio over a wide concentration range from 10 nM to 10 μM (Fig. 5a–c). N-Terminal acV$_H$H fusion ORKs provided higher maximum induction than C-terminal fusions (Fig. 5a–c). For the C-terminal acV$_H$H fusion ORKs, the variant containing the full linker domain (DcuS$_{324-543}$) showed higher

maximum induction than the ORK containing a linker domain shortened by six amino acids (DcuS$_{330-543}$) (Fig. 5b). The two-hybrid POST system that also used the acV$_H$H-DcuS$_{324-543}$ ORK, but with the two-hybrid OGR and a TetR reporter plasmid, performed with a lower maximum induction and a slightly higher dynamic range (Fig. 5c).

**Probing for possible cooperative effects**. Positive and negative cooperativity are common features of native signaling pathways and are important for bistability, ultrasensitivity, oscillations, and other complex signaling behaviors[49]. Typically, cooperativity requires the assembly of multiple proteins in such a way that ligand binding influences the binding of subsequent ligands, either positively or negatively. We speculated that the combination of N-terminal and C-terminal fusions of acV$_H$H might synergistically activate ORK dimers or facilitate ORK oligomerization. Oligomerization might enable transphosphorylation of kinase domains by neighboring protein complexes, or it could increase the local concentration of OGRs in proximity to phosphohistidine (Fig. 6a). These mechanisms could steepen the dose–response curve by providing positive cooperativity in OGR activation. We combined the N-terminal and C-terminal fusions of acV$_H$H to generate a dual-ORK ($P_{SV40}$-acV$_H$H-DcuS$_{324-543}$-acV$_H$H -pA; Fig. 6a). This construct performed with similar sensitivity but with a steeper dose–response relationship than single-acV$_H$H ORKs (Fig. 6b). The measured Hill coefficients of dual-ORK POST ranged between 0.8 and 1.5, whereas the Hill coefficients of single-ORK POST ranged between 0.4 and 0.8. Hill coefficients <1 indicate negative cooperativity of single acV$_H$H ORKs, although the mechanism might be complex[50]. The observed effect is likely influenced by the phosphatase activity of the HK and may be affected by asymmetric phosphotransfer (i.e. one phosphorylated monomer per dimer at a time, resulting in one active kinase domain per dimer) and symmetric phosphatase activity, which is a well-investigated effect in several native HKs[51]. The increase of the Hill coefficient of the dual ORK POST system compared to single ORK POST indicates that the degree of cooperativity can be amplified by the addition of an additional dimerization domain.

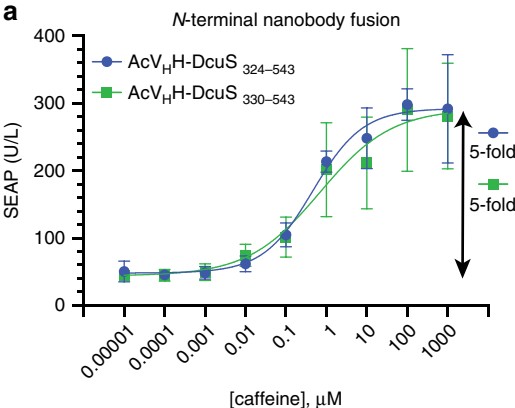

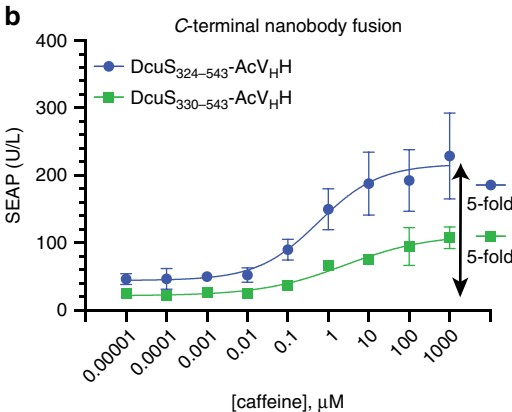

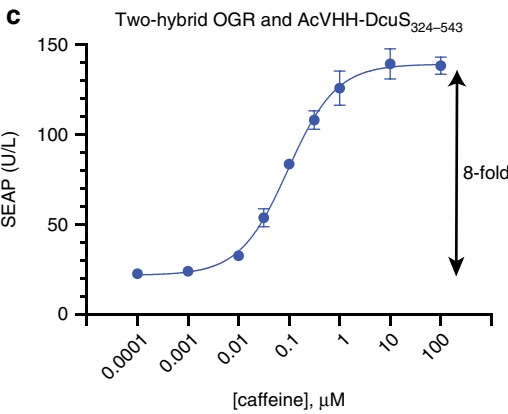

**Fig. 5 Dose–response curves. a** The dose–response curves of caffeine-induced reporter gene expression for POST with ORKs containing DcuS$_{324-543}$ or DcuS$_{330-543}$ fused to an N-terminal acV$_H$H dimerization domain. **b** The dose–response curves for ORKs with a C-terminal acV$_H$H dimerization domain. For better comparability with the performance of N-terminal acV$_H$H fusion ORKs, the Y-axis has the same scale as panel **a**. **c** Dose–response of the two-hybrid OGR design and an acV$_H$H-DcuS$_{324-543}$ ORK. Plots show the mean ± s.d. of n = 3 biologically independent samples measured 24 h after induction, and the results are representative of three independent experiments. Only error bars larger than the symbol size are displayed. Source data are provided as a Source Data file.

To test how rapidly the engineered signaling pathway can respond to caffeine stimulation, we quantified SEAP production every 2 h from cells transfected with the POST system containing the dual-ORK and the DcuR-VP16 OGR. Starting at 4 h after induction, reporter gene expression in the induced group was significantly upregulated compared to the non-induced control (Supplementary Fig. 3).

**Probing for orthogonality**. To look for possible influences of cellular contexts, we tested the POST system ($P_{SV40}$-acV$_H$H-DcuS$_{324-543}$-pA/$P_{SV40}$-DcuR-VP16-pA/DcuR-RE$_{8x}$-$P_{hCMVmin}$-SEAP-pA) in HeLa cells, bone marrow-derived immortalized mesenchymal stem/stromal cells (hMSC-TERT), and human-induced pluripotent stem cells (hiPSCs). The POST system was functional in all tested cell types and performed with a uniform dynamic range of induced vs. noninduced SEAP expression in response to caffeine (Supplementary Fig. 4a–c). Next, we examined the effect of strong activation of endogenous signaling pathways on the DcuR-operon containing reporter plasmid in cells expressing the complete POST system (ORK/OGR/reporter). KCl-induced cell depolarization and induction with TNFα led to noticeable reporter gene expression, but little or no effect was observed upon induction with forskolin, FGF, or IL-6 (Supplementary Fig. 4d). We speculate that the $P_{hCMVmin}$ promoter used in the DcuR reporter displays increased leakiness in KCl-stimulated or TNFα-stimulated cells or that the DcuR reporter contains cryptic transcription factor-binding sites for the respective endogenous transcription factors. Similar strategies as used in previous studies to reduce leaky $P_{hCMVmin}$ expression, by removing cryptic-binding sites and reducing the likelihood of long-range effects of enhancer elements[52], might reduce this effect. Finally, we investigated whether POST is orthogonal to a number of major endogenous signaling pathways. In synthetic biology, the term orthogonality is used to describe the situation where minimal interactions occur between engineered and native components of a biological system[53] and for the POST system, we primarily use this definition to indicate the low probability of POST-induced changes in endogenous signaling cascades. We cotransfected HEK-293T cells with the ORK/OGR ($P_{SV40}$-acV$_H$H-DcuS$_{324-543}$-acV$_H$H-pA/$P_{SV40}$-DcuR-VP16-pA) expression plasmids but without the OGR reporter. Instead we used endogenous pathway-specific reporter plasmids for the NF-κB, cAMP, NFAT, STAT3, and MAPK pathways and measured reporter gene expression in response to caffeine-induced ORK/OGR activation compared to known activators of respective pathways. ORK/OGR activity had no observable effect on any of the tested pathways, indicating orthogonality (Supplementary Fig. 4e–i). Further, expression or activation of the POST system had no detectable effect on the proliferation of HEK-293T cells (Supplementary Fig. 5).

**POST based on EnvZ/OmpR and NarL/NarX**. We tested if the strategy of generating POST systems by identifying minimal domains of a bacterial HK and subsequent nanobody fusion for induced dimerization can be generalized to other HKs. We focused on the well-investigated TCSs NarX/NarL, and EnvZ/OmpR[20,34]. Both TCSs have similar receptor architecture to DcuS/DcuR, but have a HAMP (present in HKs, adenylate cyclases, methyl accepting proteins, and phosphatases) domain instead of a PAS domain. For EnvZ, we based the design on a reported crystal structure of the intracellular domain (protein data bank ID: 4CTI) and generated a truncation mutant without HAMP, keeping only a conserved motif, involving the amino acids DRT, that is important for EnvZ activation[54]. We tested this construct ($P_{sv40}$_EnvZ$_{232-450}$-pA; numbering according to the full-length EnvZ; UniProt ID: P0AEJ4) in combination with a VP16 fusion of the response regulator OmpR (OmpR-OGR; $P_{sv40}$_OmpR-VP16-pA; Fig. 7a) and a reporter plasmid with OmpR response elements[34] (OmpR-RE$_{2x}$-$P_{hCMVmin}$-SEAP-pA), and observed very high basal gene expression (Supplementary Fig. 6). We speculated that the shortening of the connector domain between the DHp and CA domains could reduce the autophosphorylation activity, even in formed EnvZ dimer

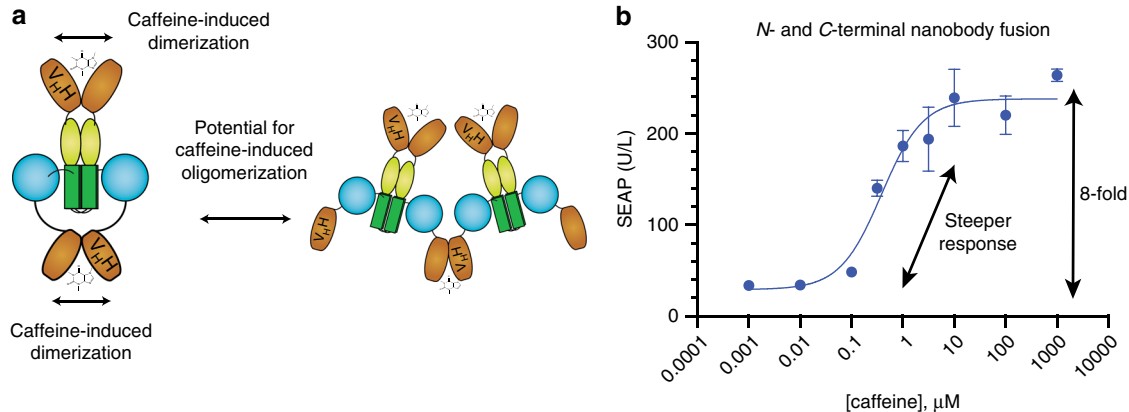

**Fig. 6 Modulating cooperativity. a** Schematic of the dual-ORK with an N-terminal and C-terminal acV$_H$H fusion. Synergistic activation of ORK dimers by acV$_H$H nanobodies or ORK oligomerization both have the potential to promote positive cooperativity. Both proposed mechanisms for the observed increase in cooperative behavior remain speculative at this point. **b** The dose–response curve of the dual-ORK. The plot shows the mean ± s.d. of $n = 3$ biologically independent samples, measured at 24 h after induction, and the results are representative of three independent experiments. Only error bars larger than the symbol size are displayed. Source data are provided as a Source Data file.

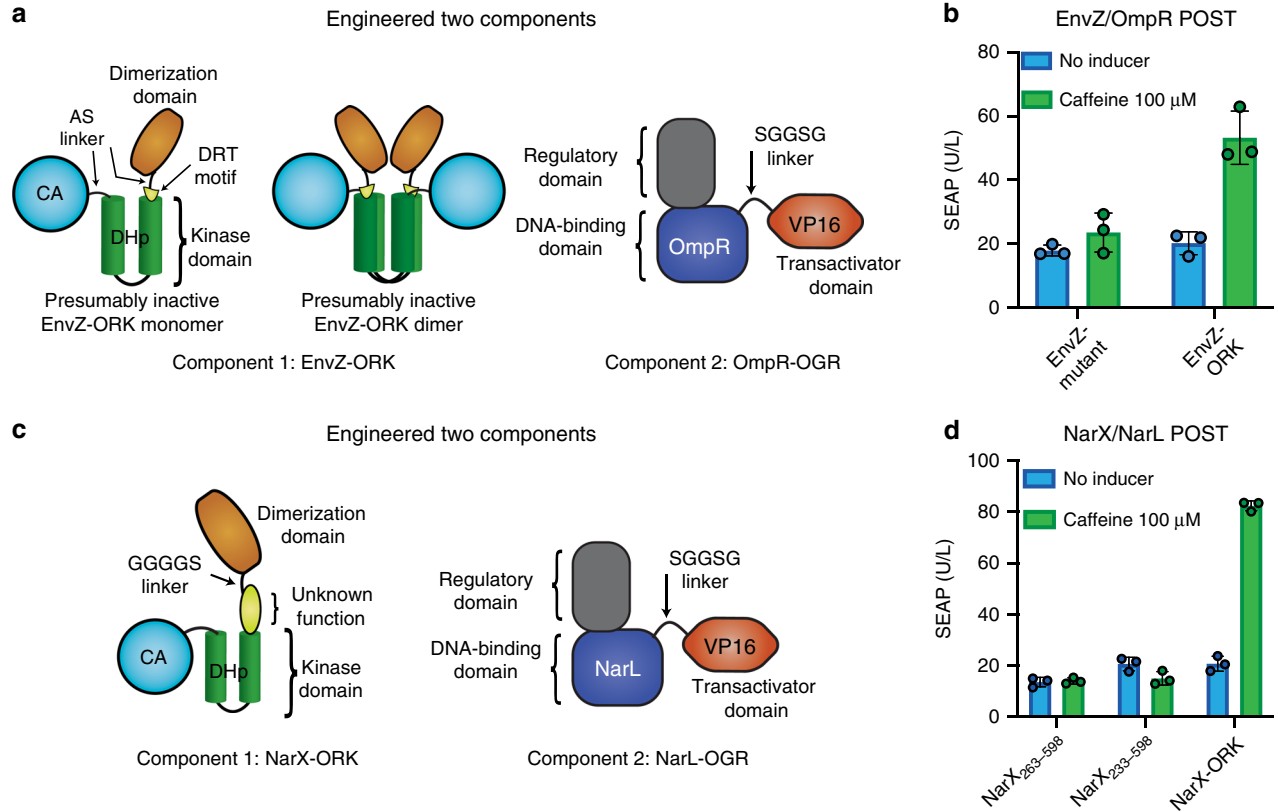

**Fig. 7 EnvZ/OmpR and NarX/NarL POST. a** Schematic of the engineered ORK/OGR proteins based on EnvZ/OmpR. In the ORK, short linkers consisting only of the two amino acids AS were used to reduce interdomain flexibility between the acV$_H$H and the kinase domain and between the CA and DHp domains by replacing the native GQEMP linker. We hypothesize that this change results in inactive dimers. **b** Induction of EnvZ/OmpR POST with caffeine. The EnvZ mutant without acV$_H$H (EnvZ$_{232-450;GQEMP:AS}$) was included as a negative control. **c** Schematic of the engineered ORK/OGR proteins based on NarX/NarL. **d** Induction of NarX/NarL POST with caffeine. The NarX truncations without acV$_H$H were included as negative controls. The bar charts show the mean ± s.d. of $n = 3$ biologically independent samples overlaid with a scatter dot plot of the original data points, measured at 24 h after induction, and the results are representative of three independent experiments. Source data are provided as a Source Data file.

complexes. In this setup, activity might be restored with acV$_H$H dimerization by the formation of functional tetramers (dimers of dimers, or higher-order oligomers) that could trans-phosphorylate each other, or by breaking the OFF-conformation of mutated EnvZ dimers. However, both proposed mechanisms would require an acV$_H$H fusion with a very

short or rigid linker domain (Fig. 7a). We generated an EnvZ mutant with short-side-chain amino acids (AS) connecting the DHp and the CA domain instead of the connector domain (GQEMP; EnvZ$_{291-295}$) to provide a minimum of flexibility and rotational freedom, but reducing its length ($P_{sv40}$-EnvZ$_{232-450;}$ $_{GQEMP:AS}$-pA). Expression of this EnvZ mutant together with the

OmpR-OGR resulted in greatly reduced activity (Fig. 7b). We fused the acV$_H$H with only a minimal (AS)-linker to EnvZ$_{232-450;}$ $_{GQEMP:AS}$ to generate an EnvZ-ORK ($P_{sv40}$-$acV_H$H-$EnvZ_{232-450;}$ $_{GQEMP:AS}$-$pA$). Expression of the EnvZ-ORK and the EnvZ-OGR resulted in caffeine-inducible gene expression from the EnvZ reporter plasmid (Fig. 7b). These results demonstrate the functionality of an EnvZ/OmpR-based POST system and add shortening of the connector domain between the DHp and CA domains to the repertoire of strategies for HK autoactivity reduction.

NarX has an additional domain with largely unknown function between the HAMP and kinase domains. We tested two truncation mutants devoid of the HAMP domain ($P_{sv40}$-$NarX_{233-598}$-$pA$; $P_{sv40}$_$NarX_{263-598}$-$pA$; NarX numbering according to the full-length NarX; UniProt ID: P0AFA2) in combination with a NarL-OGR ($P_{sv40}$-NarL-VP16-$pA$) and a reporter plasmid with NarL response elements[34] ($NarL$-$RE_{2x}$-$P_{hCMVmin}$-$SEAP$-$pA$). Neither truncation mutant was constitutively active. We chose the shorter construct to generate a NarX-ORK by N-terminal fusion to acV$_H$H. Expression of the NarX-ORK ($P_{sv40}$-$acV_H$H-$NarX_{233-598}$-$pA$; Fig. 7c) and NarX OGR resulted in caffeine-inducible gene expression from the NarL reporter plasmid, confirming that the NarL/NarX-based POST system is functional (Fig. 7d).

## Discussion

This work provides a proof-of-concept that engineered components derived from bacterial HKs and RRs can serve as modular and orthogonal signaling units in mammalian cells. In addition to employing the POST system to drive transgene expression from a DcuR-responsive synthetic operator, we show that DcuR phosphorylation causes protein dimerization. We propose that such a relay system of protein dimerization can serve as a basic building block for generating protein circuits, as demonstrated for the two-hybrid OGR design.

POST also provides a complementary method to study histidine kinase activation in the absence of the pathway interactions that can be encountered in bacteria[55]. Our results confirm the importance of the linker domain for signal transduction between the PAS and kinase domains of DcuS, in accordance with research on similar linker domains in other bacterial HKs[56,57], and suggest that the linker domain has a role beyond purely mechanical signal transduction. Reducing the linker length reduced the HK activity in all constructs, including ORKs activated by an unnatural C-terminal dimerization mechanism. These results show that even in the absence of any N-terminal sensor domain, DcuS-signaling activity in mammalian cells still requires N-terminal amino acids of the linker domain.

The presented ORKs, OGRs, and two-hybrid OGRs were developed by isolating and repurposing functional elements of the bacterial HKs and RRs to serve a predictable biological function in a non-native context. Most analyses were performed with DcuS/DcuR-derived constructs, but the functionality of EnvZ/OmpR-based and NarX/NarL-based systems suggests that the approach is generalizable. These results complement and extend other recent research[20]. POST is likely complementary to most other designs of gene or protein circuits. As exemplified with TetR, POST can be integrated into existing systems based on bacterial transcription factors. Similarly, it should be complementary to protease-based approaches for protein circuit design. Exchanging the acV$_H$H for other modules from the synthetic biology toolbox is expected to yield systems that respond to other inducers. Our findings represent a proof-of-concept of POST as a basic building block for modular phosphoregulated protein effectors, which may be the basis for controlling increasingly complex systems in mammalian synthetic biology.

POST may therefore contribute to the generation of synthetic protein circuits that approach the versatility of native signal transduction pathways and enable tightly regulated behavior of engineered cells with elaborate functions.

## Methods

**Cell culture and transfection.** Human embryonic kidney cells (HEK-293T, German Collection of Microorganisms and Cell Cultures (DSMZ): ACC 635), HeLa cells (HeLa, ATCC: CCL-2), and bone marrow-derived immortalized mesenchymal stem/stromal cells (hMSC-TERT) cells[58] were cultivated in Dulbecco's modified Eagle's medium (DMEM, 31966-021, Life Technologies Europe BV) supplemented with 10% (v/v) fetal bovine serum (FBS, F7524, lot BCBS0318V, Sigma-Aldrich) and 1% (v/v) streptomycin/penicillin (L0022, Biowest) at 37 °C in a humidified atmosphere with 5% $CO_2$ in air. For in vitro experiments, $1.4 \times 10^6$ cells in 12 mL DMEM were seeded into 96-well plates (167008, Thermo Fisher Scientific) 24 h before transfection. The transfection mix per well consisted of 125–150 ng of plasmid DNA mixed with 50 μL DMEM without any supplements and 600–900 ng of polyethylenimine (24765-1, Polysciences Inc.). The exact amounts are given in Supplementary Table 4. The transfection mix was prepared, vortexed, incubated for 20 min and then added to the inner 60 wells of the cell culture plate. Medium with the transfection mix was exchanged after 15 h for 125 μL/well medium with different concentrations of the appropriate inducer or no inducer as a negative control. Supernatant for quantification of the secreted reporter protein SEAP was taken after 24 h. The supernatant was taken from separate wells at each time point in the time course experiments.

**hIPSC preparation, culture, and transfection.** hiPSCs were derived in a previous study[59]. They originate from the adipose tissue of a 50-year-old patient using an mRNA-based reprogramming technique under feeder-free conditions. hiPSCs were cultivated in mTeSR medium (85850, Stem Cell Technologies) supplemented with Normocin 50 μg/mL (ant-nr-1, Invivogen) and cultured on Geltrex-coated 12-well plates (A1413202, Invitrogen). For transfection experiments, hiPSCs were dissociated using Accutase (A1110501, Life Technologies), and seeded at 30–50% confluency using mTESR media supplemented with 10 μM Y27632 (72302, Stem Cell Technologies) in Geltrex-coated 24-well plates. hiPSCs were transfected with 3000 ng of DNA per six wells, using a 1:3 ratio of Lipofectamine Stem transfection reagent (STEM00003, Life Technologies) according to the manufacturer's instructions. SEAP was analyzed in the supernatant 24 h later.

**Inducer preparation.** All stock solutions were prepared at 1000× concentration. 100 mM caffeine (108160100, ACROS Organics) was prepared as an aqueous solution and was stored at 4 °C for up to 4 weeks. The solution was warmed to 37 °C before experiments to dissolve precipitates. For dose–response experiments, working solutions were prepared by serial dilution in DMEM with 10% (v/v) FBS (F7524, Sigma-Aldrich) and 1% (v/v) streptomycin/penicillin (L0022, Biowest), prewarmed to 37 °C. 10 μg/mL recombinant human TNFα (300-01A, Peprotech),

Aqueous solutions of 10 μg/mL recombinant human FGF-basic (154 a.a.) (100-18B, Peprotech) and 10 μg/mL IL-6 (200—06, PeproTech) were prepared and stored at −20 °C prior to use.

A solution of 10 mM forskolin (F3917, Sigma-Aldrich) in ethanol was prepared and stored at −20 °C prior to use.

An aqueous solution of 4 M KCl (P9541, Sigma-Aldrich) was prepared and stored at 4 °C.

A solution of 10 μM rapamycin (AG-CN2-0025-C100, AdipoGen) in DMSO was prepared and stored in aliquots at −80 °C.

**Plasmid preparation.** Plasmids were generated by means of molecular cloning with the aid of restriction endonucleases (New England Biolabs, HF enzymes were used whenever possible), and ligations were done with T4 DNA ligase (M0202L, New England Biolabs). Plasmid backbones were dephosphorylated with Quick CIP phosphatase (M0508L, New England Biolabs) prior to ligation. PCRs were performed with Q5 high-fidelity DNA polymerase (M0491L, New England Biolabs). Details of all restriction enzymes and primers used for each plasmid are presented in Supplementary Table 1. The plasmids were amplified with XL10 gold chemically competent cells (C2992, New England Biolabs) and DNA was purified with a plasmid miniprep kit (D4054, Zymo Research).

**Oligo annealing.** For reporter plasmids, oligos were designed to contain operator sites and to anneal with sticky ends compatible with restriction endonuclease-based cloning. 10 μM oligo mix (5 μM each) was phosphorylated with T4 Polynucleotide Kinase (M0201S, New England Biolabs) in T4 DNA ligase buffer (M0202L, New England Biolabs) at 37 °C for 30 min. The mix was subsequently heated to 95 °C and then allowed to cool to room temperature for 15 min. A 1:250 dilution of the annealed and phosphorylated oligos was used for cloning.

**PCR amplification of bacterial genes.** The genes for DcuS and NarX intracellular domain, DcuR, NarL, and OmpR were directly PCR-amplified from XL10 gold

chemically competent cells (C2992, NEB). EnvZ intracellular domain was amplified from BL21 (C3010, NEB; BL21(DE3)pLysS-T1R) cells. 0.2 µL of bacteria were added to the PCR mix, based on the assumption that the initial denaturation step of 98 °C for 30 s is sufficient for bacterial lysis.

**SEAP measurements.** SEAP activity (U/L) in cell culture supernatants was quantified by kinetic measurements (1 measurement/min for 30 min at 37 °C) of absorbance increase due to phosphatase-mediated hydrolysis of *para*-nitrophenyl phosphate (pNPP). 4–80 µL of supernatant was adjusted with water to a final volume of 90 µL, heat-inactivated (30 min at 65 °C, then centrifuged for 1 min, 1000×g), and mixed in a 96-well dish with 100 µL of 2 × SEAP buffer (20 mM homoarginine, 1 mM MgCl₂, 21% (v/v) diethanolamine, pH 9.8) and 20 µL of substrate solution containing 20 mM pNPP (Acros Organics BVBA). Absorbance was measured at 405 nm with a Tecan Genios PRO multiplate reader (Tecan AG).

**Sequence upload to GenBank.** GenBank files and a multi-FASTA file were generated with Benchling[60]. GenBank (.gb) files were converted to feature tables with GB2Sequin[61] for sequence submission with BankIt.

**Statistics.** Data was collected with Microsoft Excel for Mac. All statistical analysis was done with GraphPad Prism 8. Graphs show the mean ± s.d. of n = 3 biologically independent samples per data point and are representative of three independent experiments. Error bars smaller than the symbol size are not shown. For Supplementary Fig. 3, we tested the statistical significance of the difference in SEAP expression at each time point, starting at 2 h post-induction. Equal variance between groups was not assumed. The Holm–Sidak method was used to correct for multiple comparisons of two-tailed *t*-tests. Exact *P* values, *t*-ratios, and degrees of freedom are provided in Supplementary Table S3.

The dose–response curves (Figs. 5 and 6) were fitted to a four-parameter logistic (4-PL) curve ([Agonist] vs. response-variable slope; Equation: $Y = \text{Bottom} + (X^{\text{Hillslope}}) \times (\text{Top-Bottom})/(X^{\text{HillSlope}} + \text{EC50}^{\text{HillSlope}})$) and the result tables including all parameters are provided in Table 2.

We follow the definition of biological replicates as "parallel measurements of biologically distinct samples"[62]. Here, this definition refers to the measurements of reporter protein activity in the supernatant of distinct wells cultured in multi-well plates under the same conditions and transfected with the same transfection mix. These controls serve to capture the effect of biological variability on the results, while all other conditions are the same. The representative graphs present the results of a single experiment under these conditions and are indicative of the qualitative results that can be obtained by replication. Technical replicates (i.e. the repeat measurement of the same sample to capture measurement and handling errors) were not performed. Independent experiments indicate the repetition of the whole experiment, including different cell passage numbers and slightly different seeding densities or DNA concentrations due to measurement and handling errors. It is therefore expected that the differences in reporter protein quantification between independent experiments are larger than those between biological replicates. Data for all figures and repeats is provided in the Source Data file.

**Viability measurement.** A Cell Counting Kit-8 (96992, Sigma-Aldrich) was used to determine cell amplification and metabolic activity. The assay involves conversion of WST-8 [2-(2-methoxy-4-nitrophenyl)-3-(4-nitrophenyl)-5-(2,4-disulfophenyl)-2H-tetrazolium, monosodium salt] by metabolically active cells to the yellow dye formazan. A 10 µL aliquot of WST-8 per well was added directly to the medium of a HEK-293T cell culture in a 96-well plate at 24 h after induction with caffeine. Absorbance at 450 nM was measured 4 h later.

**Reporting summary.** Further information on research design is available in the Nature Research Reporting Summary linked to this article.

## Data availability

All data is available in the main text or the supplementary information materials. Sequence data of original plasmids are deposited in GenBank (MT267299-MT267334) and Benchling[60]. Accession codes are provided in Supplementary Table 1. Source Data are provided with this paper.

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

## Acknowledgements

We thank Mingqi Xie and Tobias Strittmatter for valuable discussions on the manuscript, Bozhidar-Adrian Stefanov for valuable discussions on the manuscript and for support in the generation of illustrations, and Samson Lichtenstein, Sailan Shui, Sandrine Georgeon, and Stéphane Rosset for their support in experimental work. This work was supported by a European Research Council (ERC) advanced grant (ElectroGene; grant no. 785800) and in part by the National Centre of Competence in Research (NCCR) for Molecular Systems Engineering.

## Author contributions

L.S. developed the project. L.S., M.S., M.M, A.B., P.S. designed and performed the experiments. L.S., M.S., M.F. analyzed the results. L.S., M.F. wrote the manuscript.

## Competing interests

The authors declare no competing interests.

## Additional information

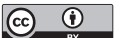

