## [Peer Review File · Nature Communications]

Reviewers' Comments:

Reviewer #1:

Remarks to the Author:

The major claim of this paper is focused on the development of functional modules that enable the integration of orthogonal phosphoregulated protein switches into the synthetic biology toolbox, for mammalian systems. To achieve this the authors repurpose components of a bacterial two-component system to create chemically induced phosphotransfer in mammalian cells. The phosphoregulated orthogonal signal transduction (POST) system is controlled by caffeine and regulates the activity of engineered kinases and transcription factors via phosphorylated histidine and aspartate residues.

The design strategy to establish the phosphoregulated orthogonal signal transduction (POST) system was based on the idea that chemically induced dimerization of the kinase domain of a bacterial histidine kinase (HK) would trigger trans-autophosphorylation of the homodimer, and this event would be followed by phosphotransfer to and activation of the corresponding response regulator (RR). The functionality of this system was tested by measuring reporter gene expression, which activates the binding of the RR fused to a transactivator domain. The phosphotransfer can be expected to be specific for the engineered HK/RR pair. Additionally, mammalian orthologues to bacterial TCSs have not been identified, and HK/RR pairs are not present in the animal kingdom. Therefore, the authors refer to the engineered kinase as the orthogonal receptor kinase (ORK) and the RR as the orthogonal gene expression regulator (OGR).

As a starting point the authors chose the prototypical homodimeric *Escherichia coli* HK DcuS, which senses C4-dicarboxylates (e.g. fumarate) and controls the RR DcuR. Design workflow: (i) identify minimal domains of DcuS that are not active when expressed cytosolically, but retain intact kinase domains; (ii) fuse these minimal domains to an anti-caffeine heavy chain nanobody (acVHH) for chemically induced dimerization; (iii) Design Goal - the generation of chemically inducible orthogonal receptor kinases (ORKs).

The findings of this paper are somewhat novel and may be of interest to the broader community – given that signal diversity can be achieved in POST systems using a similar engineering workflow moving forward. Assuming that this condition will hold, the work presented in this paper should be of interest to synthetic biologist and related investigators, and could potentially meet the criteria for publication as an article provided the authors address the following concerns raised by the reviewer.

Reviewer Comments:

1. In the introduction, please discuss other “orthogonal [bacterial] systems” used in mammalian cells, and how the POST system would complement existing “orthogonal systems”. Also, in the abstract what are the “new possibilities for protein circuit engineering in synthetic biology” that the POST system will enable?

2. The engineering workflow (i.e., “design strategy”) for the POST system is not presented effectively. The authors articulate the design strategy in lines 48-58, and summarized the purported workflow in Figure 1A. However, this section of the paper and the figure are poorly connected. Expanding this figure and reporting and labeling the anatomy of the putative POST system using the language and abbreviations in the corresponding section (i.e., lines 48-58 and lines 72-73), followed by a schema describing the intended mechanism of action would greatly improve the clarity of the design goals and strategy to achieve the desired outcomes. For example, have a figure that labels: 1) kinase domain of a bacterial histidine kinase (HK), response regulator (RR), what is regarded as the engineered kinase that corresponds to the orthogonal receptor kinase (ORK), and what corresponds to the orthogonal gene expression regulator (OGR), anti-caffeine heavy chain nanobody (acVHH), (GGGS)-linker etc.; 2) label the coloured sections of the POST system. In a separate figure, describe the putative mechanism of action for the POST system, and provide some granularity for the reporter system. In addition, the reviewer is not sure what the purpose of the “Graphical Abstract” is, it seems largely redundant (please remove). Moreover, this space could be used to improve and expand Figure 1A. Also the captions need to reflect this organization.

3. Figure 1D is too small, please make this a separate figure. I would suggest making Fig 1D-E as one figure (separate from Fig 1A-B). As the purpose of this section is to illustrate variants lacking

the PAS domain and their corresponding performances. Also, Figure 2A and the first part of Fig 2C (i.e., the first 4 data sets) overlap with Fig 1D-E. Likewise, Fig 2B and the second part of Fig 2C (i.e., the second 4 data sets) are somewhat distinct from Figure 2A and the first part of Fig 2C (i.e., the first 4 data sets). The reviewer suggest making Figure 2A and the first part of Fig 2C (i.e., the first 4 data sets) a separate figure, and , Fig 2B and the second part of Fig 2C (i.e., the second 4 data sets) a separate figure. Please articulate the key differences between the sections? Clearly discuss why the changes where made, and the relative value and importance of the corresponding results.

4. The section discussing the results summarized in Fig 2D are hard to reconcile. Graphically, what exactly is an OGR and ORK? Highlight differences that Fig 2D is conveying clearly, (i.e., relative to the previous sections) – it all looks the same. Discuss the importance of these results.

5. Please make the combined the N- and C-terminal fusions of acVHH to generate a dual-ORK a separate section in the paper, and make Fig 3D-F a separate figure. In addition, expand this section. Specifically, the authors casually make the claim that the dual system is potentially cooperative. What data supports this claim? What other experiment (and/or validation) could support this claim? Why is cooperativity important for this system (i.e., what are the advantages / disadvantages)? Could the authors have anticipated the dual system would result in cooperativity? Is this a general strategy for conferring cooperativity?

6. Lines 183 – 185 the authors state “We think that POST has the potential to serve as entry point for the design of synthetic protein circuits that approach the complexity of native signal transduction pathways.” Please rephrase this statement. To achieve protein circuit complexity on par with native signal transduction pathways will require significantly more than one POST system.

7. In general, the layout of the paper and figures are too compressed for the content. Please expand this “brief communication” to the article format with a proper introduction, result sections (with titled sections), discussion section and methods and materials section as part of the main text. In addition to breaking up the figures (and expanding the figures as indicated) please write the captions with greater detail. Also please include details with respect to the genetic materials used (i.e., all vector maps, all plasmids, all source data) as part of the supplement – i.e., sufficient details for other investigators to reproduce the results outlined in this paper.

Additional Comments:

1. It is hard to consider implications or extrapolate to the broader field of synthetic biology with only one test case. It is possible this approach only works for the specific components chosen in this work rather than as a broad tool - would need to see other combinations of HK/RR pairs with different nanobodies. Not necessary to include in this work but important to discuss the possibility and pitfalls moving forward in the discussion section.

2. Figure 3e - would like to see quantitative data on fits for dose response curves. Equations used to fit along with parameters and coefficients to demonstrate “goodness of fit”. Additionally, terms like “steeper dose response” and “signal-to-noise ratio” should be quantified and presented.

3. In supplementary Figure S1, although the dynamic range is uniform across cell types, is the difference in activity across cell types expected and/or meaningful?

4. Figure 2 - it is very hard to see the differences between the green (caffeine) and blue (no ligand) bars. Please choose other colors with more contrast.

5. Please define TCS in the main text.

Reviewer #2:

Remarks to the Author:

Scheller et al. presented the development of an orthogonal tool engineered from a bacterial two-component system (DcuSR) for phosphorylation-regulated control of protein function in mammalian cells. The reported phosphoregulated orthogonal signal transduction (POST) system, whose dimerization can be chemically induced and controlled by caffeine, mediates phosphotransfer between histidine and aspartate residues in the engineered histidine kinase

(DcuS) and the response regulator (DcuR) respectively and thus regulates the target gene expression. This work represents a significant biological engineering advancement following the previous simple implantation of the DcuSR system in mammalian cells (Hansen et al., PNAS, 2014). This reviewer is supportive of the work, but has a number of comments listed below.

Major Comments:

1. In Figure 1E, DcuS N-terminal truncation variants still showed 40-50% of the constitutive activity (DcuS203-543). Could the authors provide some insights regarding the residual DcuS activities in mammalian cells as that may become baseline leakage (noise) for a control switch?
2. In Figure 2E, the authors showed that dimerization of DcuR after phosphotransfer could activate reporter gene expression. However, to convincingly demonstrate that only dimerized DcuR could function as activated OGR, it will be good to have a DcuR control group that could not dimerize after phosphotransfer (e.g., truncate or mutate the dimerization or phosphorylation site of one DcuR).
3. In Supplemental Figure 1, the authors tested the orthogonality of POST in 3 more human cell types. The signal-to-noise ratio in those 3 cell types (only ~3-4 folds) appeared to be lower than that in HEK cells (~6-7 folds). Also HeLa appeared to have a SEAP activity level similar to HEK, with or without the inducer, while hMSC-TERT and hiPSC both had very little SEAP activity, with or without the inducer. Could the authors provide some explanations regarding these results?
4. The authors examined cross-talks between POST and major signaling pathways in mammalian cells. While POST didn't appear to have any effect on the five endogenous pathways, in Supplemental Figure 1D, KCl and TNF α inductions led to noticeable reporter activity. Was there any statistical significance when compared with the no-inducer group? Is this experiment performed with the complete POST system (or only with the DcuR-operon containing reporter plasmid)? Would that indicate any possible effect of some endogenous pathways on POST signaling (or directly on the DcuR-operator)?
5. Did the authors test the specificity of phosphotransfer with a few common mammalian phosphorylation targets to show the orthogonality of bacterial HK on endogenous mammalian proteins?
6. Did the authors try to express any other target gene with POST to further demonstrate its utility?
7. Did the authors observe or test any harmful effect or toxicity of the POST system in mammalian cells (e.g., decreased proliferation, increased apoptosis, etc.)? For biomedical applications, it is essential to learn about the toxicity of POST components that are derived from bacteria. And if there is dose-dependent toxicity, would it be possible to optimize the system to minimize the impact?
8. Is DcuSR immunogenic in mammals since they are derived from bacteria?

Minor Comments:

1. Please provide the full name of TCS in the main text.
2. In the Materials and Methods, Inducer preparation section, it will be good to specify that all inducer stock solutions were prepared at 1000X to avoid confusion.
3. In Table S2, the subtitle should be "Statistics for Figure 3F (not 3E)".
4. In Table S3, page 5, for Supplementary Figure 1b and 1c, the label "hMSC-TERT" duplicated (I assume one of the two labels should be hiPSC).
5. Some of the supplemental tables had corrupted formats that need to be fixed.
6. English proofread is recommended for a number of grammatical errors, e.g., Line 53, "....., which activates the binding....."

Print Email

Reviewers' comments:

Reviewer #1

The major claim of this paper is focused on the development of functional modules that enable the integration of orthogonal phosphoregulated protein switches into the synthetic biology toolbox, for mammalian systems. To achieve this the authors repurpose components of a bacterial two-component system to create chemically induced phosphotransfer in mammalian cells. The phosphoregulated orthogonal signal transduction (POST) system is controlled by caffeine and regulates the activity of engineered kinases and transcription factors via phosphorylated histidine and aspartate residues.

The design strategy to establish the phosphoregulated orthogonal signal transduction (POST) system was based on the idea that chemically induced dimerization of the kinase domain of a bacterial histidine kinase (HK) would trigger trans-autophosphorylation of the homodimer, and this event would be followed by phosphotransfer to and activation of the corresponding response regulator (RR). The functionality of this system was tested by measuring reporter gene expression, which activates the binding of the RR fused to a transactivator domain. The phosphotransfer can be expected to be specific for the engineered HK/RR pair. Additionally, mammalian orthologues to bacterial TCSs have not been identified, and HK/RR pairs are not present in the animal kingdom. Therefore, the authors refer to the engineered kinase as the orthogonal receptor kinase (ORK) and the RR as the orthogonal gene expression regulator (OGR).

As a starting point the authors chose the prototypical homodimeric *Escherichia coli* HK DcuS, which senses C4-dicarboxylates (e.g. fumarate) and controls the RR DcuR. Design workflow: (i) identify minimal domains of DcuS that are not active when expressed cytosolically, but retain intact kinase domains; (ii) fuse these minimal domains to an anti-caffeine heavy chain nanobody (acVHH) for chemically induced dimerization; (iii) Design Goal - the generation of chemically inducible orthogonal receptor kinases (ORKs).

The findings of this paper are somewhat novel and may be of interest to the broader community – given that signal diversity can be achieved in POST systems using a similar engineering workflow moving forward. Assuming that this condition will hold, the work presented in this paper should be of interest to synthetic biologist and related investigators, and could potentially meet the criteria for publication as an article provided the authors address the following concerns raised by the reviewer.

We thank the reviewer for his positive assessment of the manuscript and in particular for his extensive suggestions for improving the formatting and presentation of results, which have been very valuable to us in revising the manuscript. We have also conducted several additional experiments to broaden the scope of the paper.

1. In the introduction, please discuss other “orthogonal [bacterial] systems” used in mammalian cells, and how the POST system would complement existing “orthogonal systems”. Also, in the abstract what are the “new possibilities for protein circuit engineering in synthetic biology” that the POST system will enable?

We have updated the introduction and abstract along the suggested lines (e.g. “We propose a phosphoregulated relay system of protein dimerization as a basic building block for such circuits.”).

2. The engineering workflow (i.e., “design strategy”) for the POST system is not presented effectively. The authors articulate the design strategy in lines 48-58, and summarized the purported workflow in Figure 1A. However, this section of the paper and the figure are poorly connected. Expanding this figure and reporting and labeling the anatomy of the putative POST system using the language and abbreviations in the corresponding section (i.e., lines 48-58 and lines 72-73), followed by a schema describing the intended

mechanism of action would greatly improve the clarity of the design goals and strategy to achieve the desired outcomes. For example, have a figure that labels: 1) kinase domain of a bacterial histidine kinase (HK), response regulator (RR), what is regarded as the engineered kinase that corresponds to the orthogonal receptor kinase (ORK), and what corresponds to the orthogonal gene expression regulator (OGR), anti-caffeine heavy chain nanobody (acVHH), (GGGGG)-linker etc.; 2) label the coloured sections of the POST system. In a separate figure, describe the putative mechanism of action for the POST system, and provide some granularity for the reporter system. In addition, the reviewer is not sure what the purpose of the “Graphical Abstract” is, it seems largely redundant (please remove). Moreover, this space could be used to improve and expand Figure 1A. Also the captions need to reflect this organization.

Thank you for these suggestions. We have revised Figure 1 as you proposed, and we think this has minimized redundancy with the graphical abstract. We feel the graphical abstract provides a valuable visual summary for readers. However, we will remove it if the editor feels it is redundant.

3. Figure 1D is too small, please make this a separate figure. I would suggest making Fig 1D-E as one figure (separate from Fig 1A-B). As the purpose of this section is to illustrate variants lacking the PAS domain and their corresponding performances. Also, Figure 2A and the first part of Fig 2C (i.e., the first 4 data sets) overlap with Fig 1D-E. Likewise, Fig 2B and the second part of Fig 2C (i.e., the second 4 data sets) are somewhat distinct from Figure 2A and the first part of Fig 2C (i.e., the first 4 data sets). The reviewer suggest making Figure 2A and the first part of Fig 2C (i.e., the first 4 data sets) a separate figure, and , Fig 2B and the second part of Fig 2C (i.e., the second 4 data sets) a separate figure. Please articulate the key differences between the sections? Clearly discuss why the changes where made, and the relative value and importance of the corresponding results.

We agree with the reasoning and have extensively revised the figures accordingly. We also added the requested additional explanations in the text to improve clarity.

4. The section discussing the results summarized in Fig 2D are hard to reconcile. Graphically, what exactly is an OGR and ORK? Highlight differences that Fig 2D is conveying clearly, (i.e., relative to the previous sections) – it all looks the same. Discuss the importance of these results.

We have generated an additional subfigure to clarify OGRs and ORKs and added explanations in the main text.

5. Please make the combined the N- and C-terminal fusions of acVHH to generate a dual-ORK a separate section in the paper, and make Fig 3D-F a separate figure. In addition, expand this section. Specifically, the authors casually make the claim that the dual system is potentially cooperative. What data supports this claim? What other experiment (and/or validation) could support this claim? Why is cooperativity important for this system (i.e., what are the advantages / disadvantages)? Could the authors have anticipated the dual system would result in cooperativity? Is this a general strategy for conferring cooperativity?

We have generated the new figures as requested. These experiments were indeed conducted with the goal of generating positive cooperativity. Our cautious phrasing in the interpretation of the results was made in order not to over-interpret the data, as outliers in the experimental data can heavily influence the calculation of Hill coefficients in these types of biological assays. We have expanded the section as suggested and updated the methods and discussion accordingly. All the numeric results, including all parameters are provided in supplementary table 3.

6. Lines 183 – 185 the authors state “We think that POST has the potential to serve as entry point for the

design of synthetic protein circuits that approach the complexity of native signal transduction pathways.” Please rephrase this statement. To achieve protein circuit complexity on par with native signal transduction pathways will require significantly more than one POST system.

We rephrased the statement to emphasize the proof-of-concept stage of the project. We also newly include the function of two additional POST systems based on other sets of HK/RR and reporter plasmids; please refer to our response to Additional Comment 1.

7. In general, the layout of the paper and figures are too compressed for the content. Please expand this “brief communication” to the article format with a proper introduction, result sections (with titled sections), discussion section and methods and materials section as part of the main text. In addition to breaking up the figures (and expanding the figures as indicated) please write the captions with greater detail. Also please include details with respect to the genetic materials used (i.e., all vector maps, all plasmids, all source data) as part of the supplement – i.e., sufficient details for other investigators to reproduce the results outlined in this paper.

We agree and we have extensively updated the manuscript to article format as requested. Also, we updated the GenBank numbers, which link to the full vector maps of all new plasmids and will be unlocked at the same time as manuscript publication. Additionally, we provide a link to the annotated plasmid map. All source data was already submitted as a supplementary Excel file and will be available for download from the publishers’ homepage. All plasmids will be shared upon request without MTA.

Additional Comments:

1. It is hard to consider implications or extrapolate to the broader field of synthetic biology with only one test case. It is possible this approach only works for the specific components chosen in this work rather than as a broad tool - would need to see other combinations of HK/RR pairs with different nanobodies. Not necessary to include in this work but important to discuss the possibility and pitfalls moving forward in the discussion section.

We have conducted extensive additional experiments to test for broader applicability of a similar workflow. We show POST function with the established rapamycin-induced FRB/FKBP system and we also adapted two additional bacterial histidine kinases for acVHH-induced dimerization (Supplementary figure 1, Figures 5 and 6).

2. Figure 3e - would like to see quantitative data on fits for dose response curves. Equations used to fit along with parameters and coefficients to demonstrate “goodness of fit”. Additionally, terms like “steeper dose response” and “signal-to-noise ratio” should be quantified and presented.

We updated this section in the methods accordingly, updated the figures, and provide the results including all parameters in supplementary table 2.

3. In supplementary Figure S1, although the dynamic range is uniform across cell types, is the difference in activity across cell types expected and/or meaningful?

Yes, the difference in total protein production is expected and has been repeatedly observed in the past (e.g. Wang et al. Nat Biomed Eng (2018). <https://doi.org/10.1038/s41551-018-0192-3> or Chassin, et al. Nat Comm (2019). <https://doi.org/10.1038/s41467-019-09974-5>). The amount of secreted proteins is influenced by many factors, including transfection efficiency, cell density, and metabolic activity of the cell type.

4. Figure 2 - it is very hard to see the differences between the green (caffeine) and blue (no ligand) bars. Please choose other colors with more contrast.

We changed the bar graphs as requested.

5. Please define TCS in the main text.

We included a definition of TCS in the updated introduction.

Reviewer #2

Scheller et al. presented the development of an orthogonal tool engineered from a bacterial two-component system (DcuSR) for phosphorylation-regulated control of protein function in mammalian cells. The reported phosphoregulated orthogonal signal transduction (POST) system, whose dimerization can be chemically induced and controlled by caffeine, mediates phosphotransfer between histidine and aspartate residues in the engineered histidine kinase (DcuS) and the response regulator (DcuR) respectively and thus regulates the target gene expression. This work represents a significant biological engineering advancement following the previous simple implantation of the DcuSR system in mammalian cells (Hansen et al., PNAS, 2014). This reviewer is supportive of the work, but has a number of comments listed below.

We thank the reviewer for the positive assessment of our manuscript, and very helpful comments. The critical questions raised have prompted us to conduct additional experiments for controls, which have further validated our findings. Additionally, we could correct several minor inaccuracies that we would have missed otherwise.

Major Comments:

1. In Figure 1E, DcuS N-terminal truncation variants still showed 40-50% of the constitutive activity (DcuS203-543). Could the authors provide some insights regarding the residual DcuS activities in mammalian cells as that may become baseline leakage (noise) for a control switch?

We included a discussion of this point in the main text.

2. In Figure 2E, the authors showed that dimerization of DcuR after phosphotransfer could activate reporter gene expression. However, to convincingly demonstrate that only dimerized DcuR could function as activated OGR, it will be good to have a DcuR control group that could not dimerize after phosphotransfer (e.g., truncate or mutate the dimerization or phosphorylation site of one DcuR).

DcuR mutation experiments have previously confirmed that DcuR with the suggested mutation loses its ability to bind to its DNA recognition site (Janausch et al. Microbiology, 2004 <https://doi.org/10.1099/mic.0.26900-0>). We have updated the explanation and added the reference in the main text. To confirm that dimerization and not another phosphorylation-mediated effect is the primary driver of DcuR binding and reporter activation, we generated an acVHH-DcuR fusion protein (Supplementary Figure 2). The results confirm that dimerization of DcuR, without the need for phosphorylation, is sufficient to activate reporter gene expression.

3. In Supplemental Figure 1, the authors tested the orthogonality of POST in 3 more human cell types. The signal-to-noise ratio in those 3 cell types (only ~3-4 folds) appeared to be lower than that in HEK cells (~6-7 folds). Also HeLa appeared to have a SEAP activity level similar to HEK, with or without the inducer, while hMSC-TERT and hiPSC both had very little SEAP activity, with or without the inducer. Could the authors provide some explanations regarding these results?

Yes, the difference in total protein production is expected and has been repeatedly observed in the past: (e.g. Wang et al. Nat Biomed Eng (2018). <https://doi.org/10.1038/s41551-018-0192-3> or Chassin, et al. Nat Comm (2019). <https://doi.org/10.1038/s41467-019-09974-5>). The amount of secreted proteins is influenced by many factors, including transfection efficiency, cell density, and metabolic activity of the cell type. The system was initially developed and tested in HEK cells, which might explain the reduced signal-to-noise ratio in the other cell types, even though exactly the same plasmid ratios were used. Additionally, the lower transfection efficiency in these cell types likely reduced system performance overall, as a larger subset of cells will be transfected with reporter plasmids, which contribute to leakiness, but not with sufficient amounts of ORK and OGR plasmids, which are required for system function.

4. The authors examined cross-talks between POST and major signaling pathways in mammalian cells. While POST didn't appear to have any effect on the five endogenous pathways, in Supplemental Figure 1D, KCl and TNF α inductions led to noticeable reporter activity. Was there any statistical significance when compared with the no-inducer group? Is this experiment performed with the complete POST system (or only with the DcuR-operon containing reporter plasmid)? Would that indicate any possible effect of some endogenous pathways on POST signaling (or directly on the DcuR-operator)?

We included a discussion of this issue in the main text.

5. Did the authors test the specificity of phosphotransfer with a few common mammalian phosphorylation targets to show the orthogonality of bacterial HK on endogenous mammalian proteins?

We discuss this point in more detail in the main text. In particular, we emphasize that the phosphotransfer between HK and RR does not include a transphosphorylation event. Instead the RR contains a catalytic center that uses the phosphohistidine as a substrate for autophosphorylation, which makes off-target transphosphorylation events less likely. We also included this point in the abstract, in our definition of ORKs and OGRs (design strategy) and in updated figure 1 to improve the clarity.

6. Did the authors try to express any other target gene with POST to further demonstrate its utility?

We used the SEAP-reporter plasmid to provide a quantifiable readout for POST activity. We did not test other reporter proteins, but for the TETR-OGR construct used in Figure 4c, we used a TetR-responsive reporter. A similar design has been implemented previously to express different transgenes for numerous applications. Importantly, we newly include the function of two additional POST systems based on other sets of HK/RR and reporter plasmids (Figure 7).

7. Did the authors observe or test any harmful effect or toxicity of the POST system in mammalian cells (e.g., decreased proliferation, increased apoptosis, etc.)? For biomedical applications, it is essential to learn about the toxicity of POST components that are derived from bacteria. And if there is dose-dependent toxicity, would it be possible to optimize the system to minimize the impact?

To address this question, we conducted a cell proliferation assay to compare the effects of POST and a transfection control with WT cells, all in the presence or absence of caffeine, and we did not observe toxicity. The results are presented in Supplementary figure 5.

8. Is DcuSR immunogenic in mammals since they are derived from bacteria?

Most likely there are many conditions in which the answer to this question would be yes. However, as the bacterial proteins are expressed only intracellularly, the main concern with immunogenicity would be the display of bacterial peptides on MHC I proteins, which can be tolerated in some settings (Favre et al. *J Virol* (2002). DOI: 10.1128/JVI.76.22.11605-11611.2002). Additionally, there are many studies on the prediction of MHC binding motifs (e.g. Paul et al. *Clin Dev Immunol* (2013). DOI: 10.1155/2013/467852, Zeng et al. *Bioinformatics* (2019). DOI: 10.1093/bioinformatics/btz330). Such tools could be used to identify MHC binding motifs, which could be removed to reduce immunogenicity. We would like to stress that this work is at a very early proof-of-concept stage, and further optimizations will be required before potential in vivo applications can be considered.

Minor Comments:

1. Please provide the full name of TCS in the main text.

We included a definition of TCS in the updated introduction.

2. In the Materials and Methods, Inducer preparation section, it will be good to specify that all inducer stock solutions were prepared at 1000X to avoid confusion.

Thank you, we updated the section accordingly.

3. In Table S2, the subtitle should be "Statistics for Figure 3F (not 3E)".

Yes, that is correct. We updated the label.

4. In Table S3, page 5, for Supplementary Figure 1b and 1c, the label "hMSC-TERT" duplicated (I assume one of the two labels should be hiPSC).

Thank you. We corrected the label.

5. Some of the supplemental tables had corrupted formats that need to be fixed.

We assume that this remark concerns the source data, which we provided as an Excel file. The file was converted into pdf during upload, which may have caused the issue, but it will be available for download as Excel from the publisher's homepage in the final version.

6. English proofread is recommended for a number of grammatical errors, e.g., Line 53, "....., which activates the binding....."

A native-speaking scientific editor has checked the revised version.

Reviewers' Comments:

Reviewer #1:

Remarks to the Author:

The authors have addressed all of my concerns articulated in my initial review. The result is a thoughtful and well-presented manuscript describing both the design workflow and utility of a new synthetic biology tool for uses in mammalian systems. Of particular note, the expanded writing and revised figures have contributed to making this an outstanding paper – that (now) effectively communicates the key findings of this work. Given the extensive revisions, I am more or less neutral regarding the “graphical abstract”. If the authors feel that this figure is an effective summary of the work I am contented with leaving the decision between the authors and the editor.

In conclusion, the authors responses to my initial critique are superb – and allay all previous criticisms of the work. Though not explicitly requested (at least not by me) the additional extensive experiments to test for broader applicability of a similar workflow is greatly appreciated.

Minor Suggestions:

Lines 85-87: We [hypothesized] that chemically induced dimerization of the kinase domain of a bacterial histidine kinase (HK) would trigger trans-autophosphorylation of the homodimer, followed by phosphotransfer to (and dimerization of) the corresponding response regulator (RR) (Figure 1d).

Lines 170-171: We measured dose-response curves of POST systems for ORKs containing N-terminal or C-terminal fusions of the [dimerization] domain acVHH with DcuS324-543 or DcuS330-543 together with the DcuR-VP16 OGR.

Lines 177-180: The two-hybrid POST system that also [used] the acVHH-DcuS324-543 ORK, but with the two-hybrid OGR and a TetR reporter plasmid, performed with a lower maximum induction and a slightly higher dynamic range (Figure 5c).

Lines 311-313: We believe that POST offers another control option leading [to] the generation of synthetic protein circuits that approach the complexity of native signal transduction pathways.

Lines 665-667: Figure 4 caption: The two-hybrid OGR serves as [an] example of a phosphoregulated effector protein that performs its function upon dimerization. It consists of DcuR fused to TetR and DcuR fused to the DNA transactivation domain VPR.

Corey J. Wilson, Ph.D.

Associate Professor

Daniel B. Mowrey Faculty Fellow

Georgia Institute of Technology

School of Chemical & Biomolecular Engineering

Reviewer #2:

Remarks to the Author:

I am satisfied with the revisions.

Reviewers' comments:

Reviewer #1 (Remarks to the Author):

The authors have addressed all of my concerns articulated in my initial review. The result is a thoughtful and well-presented manuscript describing both the design workflow and utility of a new synthetic biology tool for uses in mammalian systems. Of particular note, the expanded writing and revised figures have contributed to making this an outstanding paper – that (now) effectively communicates the key findings of this work. Given the extensive revisions, I am more or less neutral regarding the “graphical abstract”. If the authors feel that this figure is an effective summary of the work I am contented with leaving the decision between the authors and the editor.

In conclusion, the authors responses to my initial critique are superb – and allay all previous criticisms of the work. Though not explicitly requested (at least not by me) the additional extensive experiments to test for broader applicability of a similar workflow is greatly appreciated.

Dear Prof. Wilson, thank you for your enthusiastic response. We hope the manuscript will be received as positively by the rest of the community and we thank you for your thoughtful comments and suggestions.

Minor Suggestions:

Lines 85-87: We [hypothesized] that chemically induced dimerization of the kinase domain of a bacterial histidine kinase (HK) would trigger trans- autophosphorylation of the homodimer, followed by phosphotransfer to (and dimerization of) the corresponding response regulator (RR) (Figure 1d).

Lines 170-171: We measured dose-response curves of POST systems for ORKs containing N-terminal or C-terminal fusions of the [dimerization] domain acVHH with DcuS324-543 or DcuS330-543 together with the DcuR-VP16 OGR.

Lines 177-180: The two-hybrid POST system that also [used] the acVHH-DcuS324-543 ORK, but with the two-hybrid OGR and a TetR reporter plasmid, performed with a lower maximum induction and a slightly higher dynamic range (Figure 5c).

Lines 311-313: We believe that POST offers another control option leading [to] the generation of synthetic protein circuits that approach the complexity of native signal transduction pathways.

Lines 665-667: Figure 4 caption: The two-hybrid OGR serves as [an] example of a phosphoregulated effector protein that performs its function upon dimerization. It consists of DcuR fused to TetR and DcuR fused to the DNA transactivation domain VPR.

We have done all the minor corrections.

Corey J. Wilson, Ph.D.
Associate Professor
Daniel B. Mowrey Faculty Fellow
Georgia Institute of Technology
School of Chemical & Biomolecular Engineering

Reviewer #2 (Remarks to the Author):

I am satisfied with the revisions.

Thank you for having worked with us on this manuscript.